# Beyond Strict Regulations to Achieve Environmental and Economic Health—An Optimal PM_2.5_ Mitigation Policy for Korea

**DOI:** 10.3390/ijerph17165725

**Published:** 2020-08-07

**Authors:** Kyungwon Park, Taeyeon Yoon, Changsub Shim, Eunjin Kang, Yongsuk Hong, Yoon Lee

**Affiliations:** 1Global Sustainable Development Economic Institute, Sunmoon University, Asan 31460, Korea; kngwnpark@sunmoon.ac.kr; 2Department of International Economics and Trade & Global Sustainable Development Economic Institute, Sunmoon University, Asan 31460, Korea; tay07001@sunmoon.ac.kr; 3Division for Atmospheric Environment, Climate, Air Quality and Safety Research Group, Korea Environment Institute (KEI), Sejong 30147, Korea; cshim@kei.re.kr; 4Department of Environmental Engineering, Korea University, Sejong 30019, Korea; ejkang25@korea.ac.kr (E.K.); yhong@korea.ac.kr (Y.H.)

**Keywords:** air pollutants, particulate matter (PM_2.5_), thermal power generation, auxiliary benefit, regional computable general equilibrium model

## Abstract

Growing concern about particulate matter (PM_2.5_) pressures Korea to reduce the health risks associated with its high dependency on fossil fuels. The Korean economy relies heavily on large thermal power plants—a major source of PM_2.5_ emissions. Although air quality regulations can negatively impact local economies, the Korean government announced two strict air quality mitigation policies in 2019. We develop a regional static computable general equilibrium model to simulate the economic and environmental impacts of these polices under alternative hypothetical scenarios. We separate two regions, Chungcheongnam-do, the most polluted region, and the rest of the country, in our model. As policy options, we introduce a regional development tax and a tradable market for PM emission permits, similar to an air pollution tax and a carbon permits market, respectively. The results show that allowing higher tax rates and a tradable permits market gives the optimal combination, with the PM_2.5_ emissions reduced by 2.35% without sacrificing economic growth. Since alternative options present, for example, a 0.04% loss of gross domestic product to reduce PM emissions by the same amount, our results here may present a new policy paradigm for managing air pollutants such as PM_2.5_.

## 1. Introduction

Among the adverse effects of economic growth, air quality degradation driven by fossil fuel energy sources represents a mitigation target in most countries. Air pollutants from fossil fuel power generation such as sulfur dioxide (SOx) and nitrogen oxides (NOx) caused approximately 3.7 million deaths worldwide in 2012, the largest single environmental health risk [1]. However, most economic activities in many countries rely heavily on fossil fuel energy sources, making air pollution all but inevitable. In addition, ambient particulate matter such as fine particulate matter (PM_2.5_) causes the most substantial health risks, such as cardiovascular and cerebrovascular diseases [2]. Epidemiological studies elaborate a robust relationship between long-term exposure to ambient air pollution and premature mortality. In an example going back decades, a dramatic increase in particulate matter in the 1952 “London Fog” greatly increased the infant mortality rate [3]. More recently, Apte et al. [4] addressed the mortality from air pollution and the potential health benefits of reducing ambient PM_2.5_. Aggregate effects from an aggressive global program to reduce PM_2.5_ implemented by the WHO (World Health Organization) could substantially reduce the mortality in more polluted regions, such as China and India. Since air pollutants are a transboundary problem, severe air quality degradation in China also significantly impacts the Korean peninsula [5]. Prevailing westerlies deliver air pollutants from China, exacerbating the air quality in the late-industrialized economy of the Republic of Korea (hereafter, “Korea”) [6]. Some of the literature indicates that there is a significant improvement in air quality in China recently [7,8], but air quality degradation is one of the most serious social problems that needs to be resolved in Korea.

Among the Organization for Economic Co-operation and Development (OECD) member countries, Korea has a notorious air quality level, with a mean population exposure to PM_2.5_ of 25.1 μg/m^3^ in 2017 [9]. In 2017, the Korean government launched “Comprehensive Measures for Particulate Matter Management” to lower the domestic PM_2.5_ emissions by 35% by 2024 [10]. To implement this plan, detailed action plans suggest reducing the number of coal power generation plants and restricting the operation of high-emission vehicles. The plan emphasizes the importance of controlling both primary and secondary PM_2.5_ emissions simultaneously. Thermal power plants primarily fuel late-industrialized economies like Korea, so using strict air pollutant mitigation policies to limit particulate emissions from these plants predictably drags on economic productivity. Much economic literature has attempted to predict how a reduction in air pollutants will correlate with specific policy interventions. Since a computable general equilibrium (CGE) model simulates a general equilibrium where utility is maximized through consumption and production activities with respect to major economic agents’ goals [11], it is widely used for analyzing the interaction between the production and consumption of energy-related goods and services, where ambient particulate matter is a secondary product of energy production, and where the impacts of mitigation policies can affect economic performance [12].

Nam et al. [12] showed how a CGE model can incorporate air pollutant reduction policies. Nam et al. compared a European “Clean Air Policy Package”, whose projected health benefits for the European Union more than offset the projected economic burden, to China’s need to reduce its coal dependence to meet its pollution reduction targets. PM pollution lost China 112 billion US dollars in 2005. Xiao et al. [13] examined the effects of an “air pollution tax” in China, utilizing a recursive dynamic CGE model to simulate the effects of policy options for mitigating air pollutants. The industry-specific effects were positive or negative depending on the volume of particulate matter emission. Similarly, Wu et al. [14] calculated the change in PM_2.5_ emissions resulting from trade across provinces, and concluded that more than 30% of particulate matter was caused by multiregional economic activities. Wu et al. [14] concluded that the Chinese government needs to establish different air pollutant mitigating policies that consider heterogeneous needs across stakeholders. Korea has a similar desire to reduce air pollutants, including PM_2.5_, due to its rapid economic growth fueled by a dramatic increase in energy consumption [15]. Recently, Oh et al. [6] developed a static CGE model to simulate the economic impact of air pollutant reduction policies and to test the interactions between those polices in Korea. The results show that the costs of providing clean air are high, up to 1.75% of the gross domestic product, and would cause an asymmetrical loss to high-emission over low-emission industries. To the best of the authors’ knowledge, Oh et al. [6] made the first attempt to assess the Korean air-pollutant reduction policy (i.e., “Comprehensive Measures for Particulate Matter Management”) using a CGE model. However, they assume that the government provides an emissions credit market that may be established in Korea. Xiao et al. [13] introduced an air pollution tax to high-emission industries that is similarly applied in Korea. 

In this research, we aim to find a feasible solution for ambient particulate matter reduction polices using a model that allows a mix of policies, including the simultaneous application of an “air pollution tax” and an “emissions credit trading market.” A regional static CGE model is developed to assess the PM_2.5_ emissions from multiregional business activities in Korean provinces. This general equilibrium economic system allows us to demonstrate a new equilibrium with respect to an air pollutant reduction policy scheme. We focus on Chungcheongnam-do, where thermal power plants are concentrated in Korea, and examine an optimal policy to reduce particulate matter while minimizing the negative effects on the local economy. The rest of this study proceeds as follows. Section 2 elaborates the Korean air pollutant reduction policy, including the “Comprehensive Measures for Particulate Matter Management” and the current thermal power generation situation in Korea, where most production is focused in the Chungcheongnam-do province of Korea. Section 3 discusses the methods and data. Section 4 presents scenarios and discusses the effects of policy options. Concluding remarks and further discussions are in Section 5.

## 2. Current Policy and Case Study 

### 2.1. Comprehensive Measures for Particulate Matter Management in Korea 

Recently, the Korean government has strengthened its PM mitigation policies with specific policy measures mitigating not only PM itself, but also mitigating PM precursors such as NOx, SOx, volatile organic compounds (VOCs), and ammonia that contribute to secondary PM production [15]. The Korean government first set mitigation goals for PM precursors in the “Comprehensive Measures for Particulate Matter Management” in 2019. The mitigation goal has since been tightened to a national PM_2.5_ goal of 16 μg/m^3^ or below, with reductions to domestic emissions of 35% by 2024, through “The Special Law for Mitigating Particulate Matter (PM) Pollution” in 2019 [16]. That legislation defined the Korean PM pollution as a “social disaster” and greatly increased the national budget for urgent PM control policies, raising the budget by about 1.6 billion USD in 2019. In order to make this policy successful, the Korean government needs to prepare effective policy tools to control diverse domestic emission sources across the transportation, energy, and agriculture industries. In particular, Chungcheongnam-do (hereafter, CN) is known to have higher PM pollution due to its complex emission sources, including industry complexes (e.g., the Daesan industry complex), large coal power plants on its western coast (at Dangjin and Taeahn), and non-point emissions from agricultural areas (i.e., ammonia emissions from the livestock industry); see Figure 1 [17]. Thus, the Korean government has designated an additional three selected air pollution management areas that have higher emissions with higher annual PM concentrations in Korea. For instance, Chungcheongnam-do, one of the air pollution management areas, has to provide a regional plan for emission reductions, and its implementation actions need to be reviewed and evaluated regularly by the Ministry of Environment in Korea. These new regional targets are based on the recent “Basic Plan of Air Quality Management for Regional Air Pollution Management Area” [18]. This new regional strategy is very crucial for meeting the national PM reduction targets from the Comprehensive Measures for Particulate Matter Management, announced in 2019.

### 2.2. Case Study: Chungcheongnam-Do and Korea

It is well known that fossil-fuel power plants are major sources of air pollutants, including particulate matter. For instance, coal-fired power plants and natural-gas-fired power plants emit SOx and NOx, including PM [19]. In Korea, there are four different types of power plants unevenly distributed across the nation. The Korean peninsula is geologically high in the east and low in the west, with most of the hydro-power plants located in the eastern parts of Korea (Gangwon-do), where ample natural fresh water flows most of the year, fed by mountain rains. Nuclear power plants need a massive water supply for cooling and large industrial complexes to insure the effectiveness of the generation and transmission of electricity. They are mostly located in the southern part of the peninsula (e.g., Gyeonsangbuk-do and Jeollanam-do). To meet peak-load electricity demand, thermal power plants are built in inland areas such as CN. The annual power generation in Korea is approximately 520,000 GWh, of which about 75% is supplied by thermal power plants. Of those thermal power plants, approximately 30% are located in CN, which has been experiencing severe air quality degradation. CN is responsible for about 22% of the national annual power generation, mostly from thermal power plants; see details in Figure 2 and Appendix A, Table A1 [20]. Moreover, industrial complexes (i.e., Daesan industry complex) and a manufacturing-based regional industrial geography may exacerbate the air conditions in CN. In terms of gross domestic product (GDP), the gross regional domestic product (GRDP) of CN is only about 6% [21] of the national total. 

With a high concentration of thermal power plants in contrast to its low contribution to the GDP, CN’s annual emissions of PM_10_ and PM_2.5_ are almost 25% of the annual national emissions, accounting for approximately 31,000 tons of PM_10_ per year and 76,000 tons of PM_2.5_ per year of the Korea-wide totals [22]. Most of the PM_10_ and PM_2.5_ emissions come from major emission industries: agriculture, forestry, and fisheries; petroleum and coal; chemicals; non-metallic minerals; basic metals; electricity, gas, and steam supply; construction; and transportation. Of those industries, the basic metals industry is the largest PM polluter in CN, with more than 80% of the total PM emissions for CN. Non-metallic mineral products and transportation are the biggest PM-emitting industry sectors in Korea outside of CN, accounting for 96% of the PM_10_ and 97% of the PM_2.5_ total emissions in Korea, respectively. The PM emission contribution by industry in CN and the rest of the Korean provinces (hereafter, RP) can be found in Figure 3.

According to Kim et al. [17], CN experiences severe air quality degradation, coupled with a high density of PM-polluting industries and large thermal power plants that are treated as large point-sources. Figure 4 clearly shows that CN suffers from almost year-round PM pollution.

## 3. Methodology and Data

### 3.1. Regional Computable General Equilibrium Model

Most air quality regulations have negative impacts on the economy in general. Respecting this, we developed a single-country regional static computable general equilibrium (CGE) model to analyze the economic effects associated with policies regulating air quality. Based on a traditional neoclassical CGE model [23], a single-country economy is divided into two parts, the CN and the remaining Korean provinces (RP), to compare the regional effects using a regional input–output table. Two policy mechanisms for controlling PM emissions are allowed in the model—a PM_2.5_ emission permits market and the pollution tax rates. The CGE model divides the economy into production, household, government, savings and investment, and overseas sectors. The model consists of balanced equations that represent each sector’s optimal behavior and the flow of goods and production factors between sectors. In Figure 5, the production sector comprises a two-stage nesting structure. At the final level, domestic production is a composite of the value added and the intermediate input from Leontief technology. At the bottom level, capital and labor produce a composite of the value added in a constant elasticity of substitute (CES) technology.

In the production sector, the demand functions for the input factor demand are derived through the producer’s cost minimization as Equations (1) and (2).
(1)$$Min\;C=(1+Tl_{i})\cdot P_{L}\cdot L_{i}+(1+Tk_{i})\cdot P_{K}\cdot K_{i},$$
(2)$$s.t.\;XD_{i}=aF_{2}{\left(\gamma F_{2}\cdot K_{i}^{-\sigma F_{2}}+(1-\gamma F_{2})\cdot L_{i}^{-\sigma F_{2}}\right)}^{-\frac{1}{\sigma F_{2}}},$$
where C is the cost function; $XD_{i}$ is the final domestic production of i industry; $L_{i}$ is the labor input; $K_{i}$ is the capital input; $P_{L}$ is the labor price; $P_{K}$ is the capital price; $Tl_{i}$ is the labor tax rate; $TK_{i}$ is the capital tax rate; $aF_{2}$ is the technical efficiency parameter; $\gamma F_{2}$ is the distribution parameter; and $\sigma F_{2}$ is the elasticity of substitution.

Various measures are possible for mitigating pollutants generated as by-products in the production process: direct regulation, an emissions tax, and a tradable permits market. The implementation of an emissions tax is practically difficult when the source of a pollutant is hard to identify. On the other hand, it is easier to measure outputs than emissions, so a tax on production can induce industries to reduce output and polluting emissions without additional information. When the tax imposed on the final product is increased, the zero-profit condition at the tax rate changes, but the production function remains unchanged in the CGE model. If, to reduce air pollutants, a tax is imposed in proportion to the final domestic production, the imposed ${\mathrm{tax}}^{*}$ is added into the zero-profit condition at the final level of the nested production model, which is shown in Equation (3).
(3)$$PD_{i}\cdot XD_{i}\cdot (1-tax^{*})=(1+Tk)\cdot P_{k}\cdot K_{i}+(1+Tl_{i})\cdot P_{L}\cdot L_{i}+\left(\sum_{j=1}^{J}P_{j}\cdot io_{ij}\right)XD_{i},$$
where $PD_{i}$ is the domestic price of i industry, $P_{j}$ is the domestic price of other product of industry j, and io is the coefficient of intermediates inputs in the input-output table.

A system of marketable emissions permits is designed to induce the efficient use of resources by properly incentivizing polluters. An emissions trading system directly controls quantity, while an emissions tax controls price. Polluters can buy and sell their own permits to emit pollutants, and their emissions permits can be traded in the market. To ensure that the production cost increases by the number of permits purchased, air pollution permits are issued proportional to the amount of PM_2.5_ emissions that the firm releases in its (fixed-technology) production process. The free allocation of permits is modeled so that the firm receives subsidies from the government equal to the cost of purchasing the freely allocated permits. Therefore, as the tradable market is introduced, the company’s production function remains unchanged, but its cost function (Equation (4)) and zero-profit condition (Equation (5)) change.
(4)$$Cost=(1+Tl_{i})P_{L}\cdot L_{i}+(1+Tk_{i})P_{K}\cdot K_{i}+PPM_{2.5}\cdot ePM_{2.5},$$
where $PPM_{2.5}$ is the price of PM_2.5_ and $ePM_{2.5}$ is the quantity of the PM_2.5_ emissions.
(5)$$PD_{i}\cdot XD_{i}\cdot (1-tax^{*})=(1+Tk)\cdot P_{k}\cdot K_{i}+(1+Tl_{i})\cdot P_{L}\cdot L_{i}+\left(\sum_{j=1}^{J}P_{j}\cdot io_{ij}\right)XD_{i}+PPM_{2.5}\cdot ePM_{2.5}$$

The household sector represents all consumers, as suppliers of labor and capital for consumption, savings, and taxes. In addition, household income consists of labor and capital supply and government transfers. The demand functions for consumer goods are derived through the process of utility maximization under budget constraints, and a Cobb–Douglas function represents the household utility function in the model.

The model assumes that governments exist in the CN and the RP regions. The additional tax revenue from the air pollution reduction policy accrues to each local government and is assumed to be applied efficiently to offset damages—that is, for ecological conservation and the health of local residents. Government revenue consists of taxes on consumption, taxes on production, income taxes, and tariffs between the two regions. Similarly, government expenditures are the government consumption of consumer goods, labor and capital, and household transfers. The government converts the remaining amount (revenues minus expenditures) into savings. The demand functions for government consumption are derived through the process of utility maximization, again using a Cobb–Douglas utility function. Additional tax revenue is levied on thermal power generation in each region.

The total savings consist of household savings, government savings, and foreign savings. Total savings are invested by each industry, with each industry’s investment demand function derived through a utility maximization problem within its total savings. This study assumes that a Cobb–Douglas function is the utility function for the investment sector. The foreign sector is consumed domestically or exported with constant elasticity of transformation (CET) technology under the Armington assumption. The model structure can be found in Figure 6.

### 3.2. Sector Classification

Regional CGE models are mainly used to analyze regional economies, economic development, and policy options. CGE models are built either by applying Armington’s [24] assumptions about transactions among regions, or by incorporating a complete CGE model for each region as if the total national economy were a global model [25]. In this research, a regional CGE model was constructed using an industrial classification to establish a framework for a regional input–output table and to exclude unnecessary assumptions. A social accounting matrix (SAM) defines transactions among industries and other agents and is used to calibrate the CGE model. The calibrating SAM for this project is based on a Bank of Korea regional input–output table [26], with 2013 taken as the base year. We classify 60 industries (i.e., 30 industries, 2 regions), with industries in CN between 1 and 30, and those in RP between 31 and 60. The elasticity of substitution between labor and capital at the stage of value-added production is applied the same, regardless of region. Details of the industry classification and PM emissions can be found in Table A2.

In this research, the PM_2.5_ emissions in the base year are classified by the same SAM criteria in order to analyze changes in the PM_2.5_ emissions. While carbon emissions are released in proportion to fossil-fuel combustion, PM_2.5_ emissions are less likely to be highly correlated with CO_2_ emissions, so air pollutant emissions are defined in proportion to the industrial final production. Air pollutant emissions per value of domestic production (g/USD) are obtained by dividing each quantity of emissions in grams by the regional domestic production in USD. Thus, each industry has its own emissions coefficient for PM_2.5_. The coefficients represent the emission intensity, and are calculated by pairing the national air pollutant emission figures [22] with the regional input–output table industries [26,27]. Figure 7 shows that the emissions from basic metal products and transportation are particularly high in CN, while non-metallic products and transportation are high in RP (see Appendix A, Table A2, for more details).

## 4. Scenario Building and Results

### 4.1. Scenarios

It is well known that an emissions tax is more efficient and effective than most means of direct regulation in mitigating air pollution. However, a direct regulation of pollutants may be more cost-effective to implement. In fact, the main policy schemes in Korea for mitigating air pollution of the PM type are direct regulations. For instance, the Korean government announced regulation policy measures to reach national air standards by shutting down aging coal-fired power plants and enforcing upper limits to power generation when the concentrations of PM_10_ and PM_2.5_ are particularly high [15]. To analyze the policy effects on the Korean economy, the aforementioned policy schemes can be treated as spontaneous events (external shocks) to an economic structure. In this way, our experiment is focused on the long-term economic effects and environmental impacts of air quality regulation—e.g., in response to a regional development tax in Korea. The purpose of this tax, imposed as a proportion of power generation, is to secure funds for balanced regional development, improvements to environmental quality, and the protection of water resources.

The current tax is 0.03 cents per kWh for thermal power, 0.2 cents per kWh for hydro power, and 0.1 cents per kWh for nuclear power. However, there has been increasing pressure to raise the tax rates for thermal power, equivalent at least to the level of water or nuclear power. The revenue from this tax could be used to improve the pollution mitigation capacities of thermal power plants, lowering emissions that affect local ecosystems and local residents [28]. If this regional development tax rate increases to the nuclear power rate, the long-term economic and environmental impacts could be more permanent. With this in mind, we simulate the permanent impacts on regional economies and the environment by executing a sensitivity analysis, ranging the tax from the status quo (0.03 cents per kWh) up to 1 cent per kWh. In the scenarios, the regional development tax impacts the electricity, gas, and steam supply industry in Table 1, which is industry 16 for CN and industry 46 for RP, and the ratio of the tax is recalculated for each scenario.

In addition, an emission permits market is designed to provide economic incentives to polluters, similar to an emissions tax, an emissions charge, or an emissions trading system to control the quantity of pollutants [29]. In this research, we introduce a hypothetical PM emission permits market system to analyze whether this policy scheme can reduce the particulate matter in Korea or not, and to analyze what the total economic impact will be, whether negative or positive. In total, we provide seven different combinations of scenarios to calculate the regional economic impacts, and from this set determine the optimal policy for reducing PM in Korea. Scenarios used in this research can be found in Table 2.

### 4.2. Economic Impact

Under the current regional development tax scheme, the GDP in Korea is approximately 1303 billion USD. Of this number, CN accounts for 81 billion USD, and RP accounts for 1222 billion USD. Although the national GDP increases gradually as the regional development tax rises, the GRDP in CN slightly decreases by 0.04% under the alternative “tax rate as for water power plant” scenario—i.e., 0.2 cents/kWh without introducing a permits market system. In this scenario, the government expenditure increases by approximately 2.05%, but household consumption and investment decrease 0.17% and 0.07%, respectively. It is assumed that government expenditure is not enough to compensate the consumption decreases. Moving from the “moderate” to the “strong” regulation scenario—i.e., imposing a 0.5 cent/kWh and 1 cent/kWh tax still without a permits market—the GRDP increases for CN and decreases for RP. However, the increase in the CN GRDP more than offsets the decrease in the RP GRDP, leading us approximately 1303 billion USD in the national GDP, a 0.24% increment compared to the status quo.

When a PM_2.5_ emission permits market is introduced in the model, the government revenue increases, and the new revenue is assumed to be reinvested in the clean energy sector locally. In the last of the seven scenarios, “stro + M (strong regulation with permits market),” we discover the optimal result, maximizing the aggregate economic impact nationally at 1303.26 USD. The 81.39 billion USD for the CN is the highest across the scenarios at +0.48% over the status quo, more than enough to offset a −0.028% loss in the GRDP for RP. Table 3 presents a detailed comparison of the GRDP changes under each scenario for CN and for RP.

### 4.3. Environment Impact

The PM_2.5_ emissions in the CGE model are calculated by multiplying the emission intensity by the total production. As the tax rate increases from 0.2 cents to 1 cent, the national total emissions decrease from 0.1% to 0.64%, compared to emissions at the current tax rate. There is a difference between CN and RP in the reduction pattern with respect to industry classification. The top five PM_2.5_-emitting industries (basic metal products; electricity, gas, and steam supply; construction; transportation; and non-metallic mineral products) were defined as high-emission industries under the status quo. In the current situation, the top five industries account for 98% of the PM_2.5_ emissions in CN and 91% in RP. The decrease in CN is larger than that in RP, and these regional differences were especially noticeable in the five high-emission industries. With the tax rate rising from 0.2 cents to 1 cent, the emissions in CN decrease between 0.19% and 1.11%, while they decrease only between 0.08% and 0.50% in RP. The reductions for the five high-emission industries are greater than the reduction for the other 25 industries in both CN and RP in every scenario.

The introduction of a PM permits market has a positive effect on the environmental performance when combined with an increase in the tax rate. As the tax rate increases from 0.2 cents to 1 cent, the national total emissions decrease from 0.44% to 2.35% with a tradable market, specifically from 0.65% to 3.53% in CN and from 0.37% to 1.99% in RP. The presence of an emissions market varies depending on the industry clustering in both CN and RP. In the case of the five high-emissions industries, the reduction is about three times in CN and about four times in RP compared to the status quo scenario. The reduction for the remaining 25 industries due to the presence of the market is relatively small. Compared to Oh et al. [6], the costs of reducing the PM air pollution are relatively small in our simulation. A comparison between CN and RP in terms of PM_2.5_ emissions under different scenarios is shown in Table 4.

## 5. Conclusions

Among growing concerns about environmental problems, an economy with a high dependency on fossil-fuel energy sources faces degraded air quality, especially due to fine particulate matter (PM_2.5_) emissions. Recently, clinical pathology research has demonstrated that frequent exposure to PM_2.5_ aggravates health risks, such as cardiovascular diseases and respiratory diseases. PM_2.5_ is emitted from energy sources such as thermal power plants that heavily rely on fossil-fuel energy sources to generate electricity. These are favored by rapidly growing economies because they are relatively cheap compared to renewable energy sources, such as solar power, wind, etc. However, PM_2.5_ emissions are known to decrease labor productivity and increase health expenditures—both effects that can worsen the gross domestic product (GDP). Since Korea has achieved rapid economic growth by relying on electricity from thermal power plants, deteriorating air quality in Chungcheongnam-do (CN), where large thermal power plants and large industrial complexes are located, has been inevitable. As a consequence, the Korean government announced two strict air quality improvement policies in 2019 for mitigating the PM_2.5_: “Comprehensive Measures for Particulate Matter Management,” and “The Special Law for Mitigating Particulate Matter (PM) Pollution.” These policies contain significant action plans, including shutting down aging coal-fired power plants and establishing PM mitigation targets. These implementation plans may predictably cause negative ripple effects through local economies, particularly where large thermal power plants are located, as in CN.

This research presents a single-country regional static computable general equilibrium (CGE) model for simulating the effects of PM mitigation polices under seven alternative hypothetical scenarios. These scenarios are less aggressively and more economically efficient than shutting down aging power plants and forcing generators to stop. The model allows us to navigate the optimal combined solution for mitigating PM, in economic and environmental terms, by internalizing PM_2.5_ emission burdens within the market and regulatory (pricing and tax) system. CN in Korea was selected as a representative case study due to its high dependency on thermal power plants and industrial complexes. Instead of a severe PM mitigation action plan such as shutting down power plants, two economically flexible policy experiments were simulated: a regional development tax imposed as a proportion of power generation, and a tradable PM_2.5_ emission permits market similar to a carbon trading market. Four different regional development tax rates are considered as scenarios: 0.03, 0.2, 0.5 and 1 cent per kilowatt hour. These tax rates are applied across the entire nation, and the tax revenues in CN are greater than those in RP, due to CN’s high concentration of thermal power plants. The model assumes that the new government tax revenue is reinvested in clean energy sectors. Owing in part to this effect, the GRDP in CN increases when the Korean government imposes higher tax rates on fossil-fuel energy production. Our simulation results offer a superior solution when a tradable PM_2.5_ emissions market is introduced simultaneously. From an environmental perspective, the aggregate PM_2.5_ emission reduction in the country is approximately 74,000 tons annually, and high PM emission industries in CN have to reduce their PM emission dramatically compared to those in RP under the optimal solution. This simulated number could be converted into an auxiliary benefit—a reduction in health risk.

We believe that by navigating hypothetical combined solutions, this research provides useful alternative policy schemes to reform ambient air quality, but concede that several modifications to our model could improve future investigations. First, introducing the other main sources of PM emissions, such as industrial complexes and the agricultural sector, would better reflect reality. Further, diverse emission source dynamics could introduce a new dimension beyond what was done in this research. Second, the secondary PM components and ammonia, other important byproducts of fossil-fuel energy sources, are ignored to simplify this model. Considering the biogeochemical reaction of ammonia in the atmosphere may lead us to a different policy path for mitigating air pollutants. Third, this study does not calculate the indirect costs of auxiliary benefits from reducing health risks linked to PM emissions, hence a formal cost-benefit analysis of implementing air quality improvement plans could lead to a new optimum economic equilibrium. Finally, since particulate matter can be treated as a transboundary pollutant, a broader research area including at least northeast Asia should be investigated to better simulate real-world conditions. We believe that our first attempt to study mitigation policies for air pollutants by introducing inter-regional economic differences is a good starting point and provides some guidance for how to steer promising research in the future.

## Figures and Tables

**Figure 1 ijerph-17-05725-f001:**
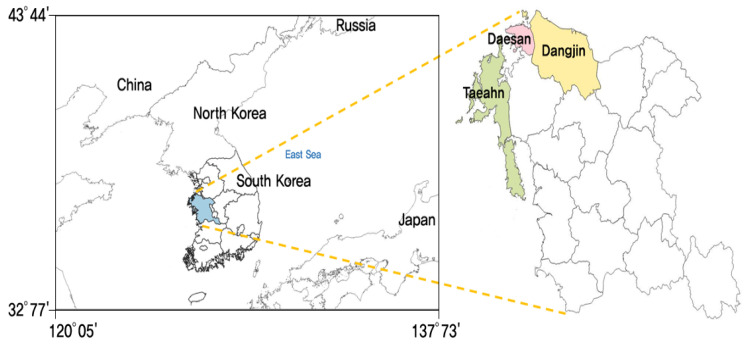
Location of large point-source particulate matter (PM) pollutants (i.e., Daesan industry complex) and coal-power plants (at Dangjin and Taeahn) in Chungcheongnoam-do (CN).

**Figure 2 ijerph-17-05725-f002:**
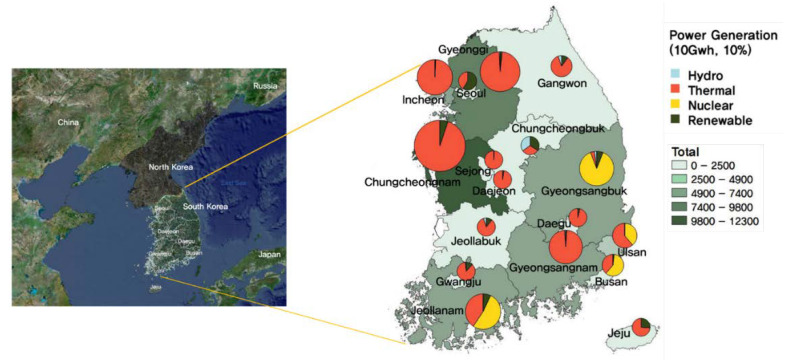
Annual power generation by regions and sources in Korea (note: circle size indicates the relative total amount of power generation when the value of CN is equivalent to 1).

**Figure 3 ijerph-17-05725-f003:**
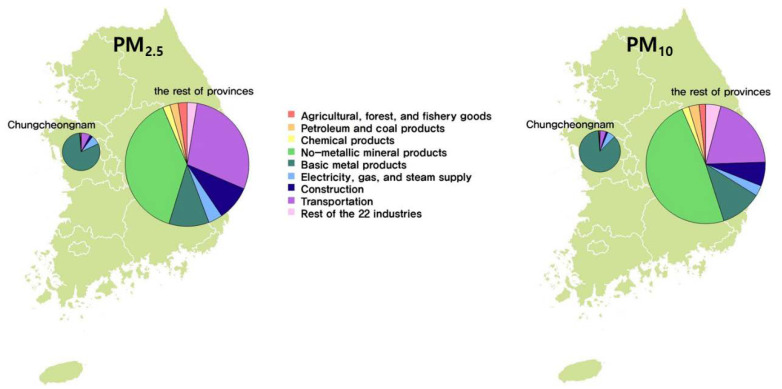
Comparison between Chungcheongnam-do (CN) and the rest of the Korean provinces (RP) for the high-emission industries of PM_2.5_ (left) and PM_10_ (right) in 2014 (note: circle size indicates the relative total amount of power generation when the value of CN is equivalent to 1).

**Figure 4 ijerph-17-05725-f004:**
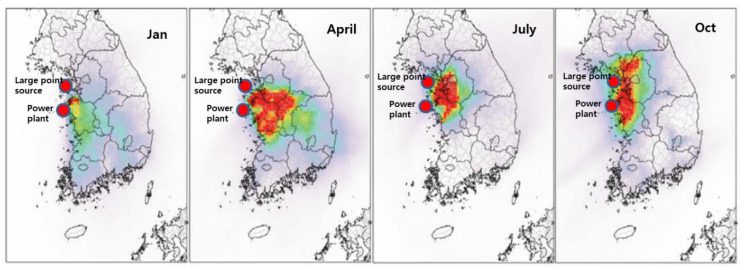
Spatial distributions of the monthly averaged contributions of large point-sources on the monthly PM_2.5_ concentration in Chungcheongnam-do (CN), modified from [19] (note: reference year is 2014).

**Figure 5 ijerph-17-05725-f005:**
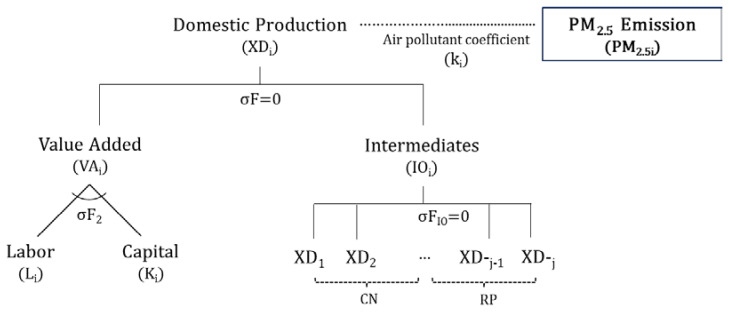
Nesting structure of the production sector in a single-country regional static computable general equilibrium (CGE) model.

**Figure 6 ijerph-17-05725-f006:**
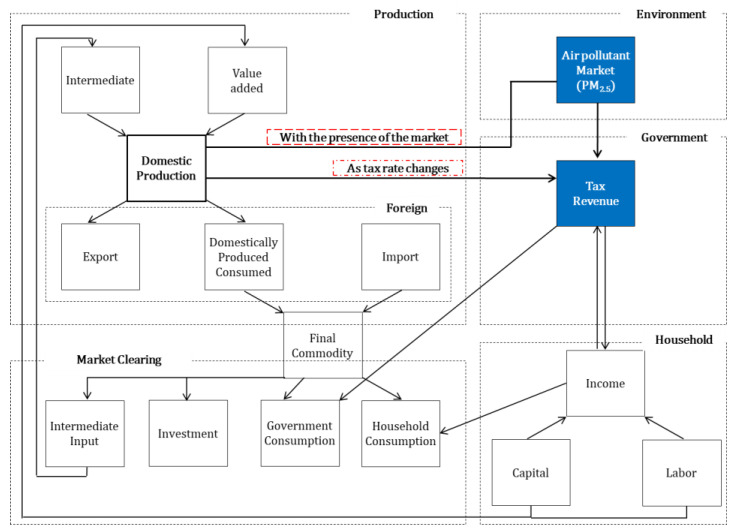
Structure and interwoven relationships among the agencies of the regional computable general equilibrium model (note: blue squares are two policy schemes used in the model and arrows indicate the flow of economic activities).

**Figure 7 ijerph-17-05725-f007:**
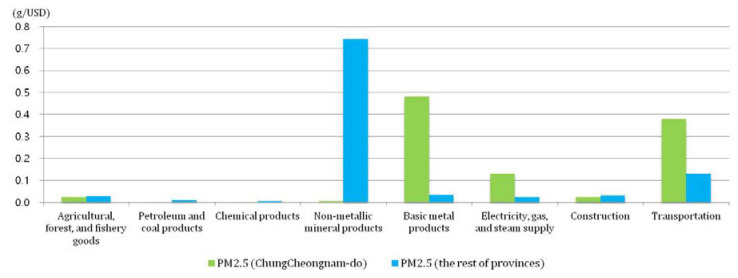
PM_2.5_ emission per domestic production for the 8 highest emission industries in CN and RP.

**Table 1 ijerph-17-05725-t001:** Industry classification, modified from [26,27].

Industry Classification		$\sigma {F_{2}}^{1}$	Industry Classification		$\sigma {F_{2}}^{1}$
1; 31	Agricultural, forestry, and fishery goods	1.014	16; 46	Electricity, gas, and steam supply	1.004
2; 32	Mined and quarried goods	0.902	17; 47	Water supply, sewage, and waste management	0.992
3; 33	Food, beverages, and tobacco products	0.995	18; 48	Construction	0.992
4; 34	Textile and leather products	1.005	19; 49	Wholesale and retail trade	0.999
5; 35	Wood and paper products, printing, and reproduction of recorded media	0.998	20; 50	Transportation	1.009
6; 36	Petroleum and coal products	1.101	21; 51	Food services and accommodation	0.971
7; 37	Chemical products	0.999	22; 52	Communications and broadcasting	1.01
8; 38	Non-metallic mineral products	0.998	23; 53	Finance and insurance	1.004
9; 39	Basic metal products	0.976	24; 54	Real estate and leasing	0.931
10; 40	Fabricated metal products, except machinery, and furniture	1.004	25; 55	Professional, scientific, and technical services	0.995
11; 41	Machinery and equipment	0.985	26; 56	Business support services	1.011
12; 42	Electronic and electrical equipment	1.001	27; 57	Public administration and defense	1.001
13; 43	Precision instruments	1.001	28; 58	Educational services	1.002
14; 44	Transportation equipment	0.985	29; 59	Health and social work	0.987
15; 45	Other manufactured products and outsourcing services	1.002	30; 60	Cultural and other services	0.999

^1^ Elasticity of substitution.

**Table 2 ijerph-17-05725-t002:** Scenarios for combined air quality improvement policies.

Scenario	Description	Tax Rate (Cent/kWh)	Permit Market
Status quo (baseline)	Current situation	0.03	No
Current + M	Implementing permit market only	0.03	Yes
Hyd	Applying regional development tax of hydropower	0.20	No
Hyd + M	Applying regional development tax of hydropower and implementing permit market simultaneously	0.20	Yes
Mod	Applying regional development tax higher than hydropower	0.50	No
Mod + M	Applying regional development tax higher than hydropower and implementing permit market simultaneously	0.50	Yes
Stro	Applying regional development tax significantly higher than hydropower	1.00	No
Stro + M	Applying regional development tax significantly higher than hydropower and implementing permit market simultaneously	1.00	Yes

**Table 3 ijerph-17-05725-t003:** Comparison of GRDP under different scenarios for CN and RP.

Scenario	Status Quo	Hyd	Mod	Stro	Hyd + M	Mod + M	Stro + M
National GDP (billion USD)	1303.20	1303.21	1303.22	1303.24	1303.22	1303.24	1303.26
GRDP in CN (billion USD)	81.00	80.96	81.02	81.20	81.12	81.21	81.39
% change in GRDP	N/A	−0.04	+0.03	+0.24	+0.15	+0.27	+0.48
% change in household		−0.17	−0.48	−1.01	−0.21	−0.56	−1.18
% change in government		+2.05	+5.92	+13.03	+3.07	+8.29	+18.15
% change in investment		−0.07	−0.20	−0.43	−0.13	−0.35	−0.77
GRDP in RP (billion USD)	1222.21	1222.24	1222.20	1222.04	1222.10	1222.02	1221.87
% change in GRDP	NA	+0.0032	−0.0004	−0.0135	−0.0088	−0.0150	−0.0277
% change in household		−0.08	−0.22	−0.48	−0.10	−0.27	−0.58
% change in government		+0.31	+0.86	+1.80	+0.44	+1.18	+2.51
% change in investment		−0.07	−0.20	−0.43	−0.13	−0.35	−0.76

**Table 4 ijerph-17-05725-t004:** Comparison between CN and RP in terms of PM_2.5_ emissions under different scenarios.

Scenario	Status Quo	Hyd	Mod	Stro	Hyd + M	Mod + M	Stro + M
National emissions (tons)	75,565	75,486	75,339	75,081	75,234	74,741	73,791
Total % changes		−0.10	−0.30	−0.64	−0.44	−1.09	−2.35
Emissions in CN (tons)	17,626	17,593	17,534	17,431	17,511	17,336	17,004
Total % change		−0.19	−0.52	−1.11	−0.65	−1.64	−3.53
% change in 5 high-emission industries	17,383	−0.19	−0.53	−1.12	−0.66	−1.67	−3.58
% change in the 25 other industries	243	−0.05	−0.12	−0.24	−0.01	−0.07	−0.16
Emissions in RP (tons)	57,939	57,893	57,805	57,650	57,724	57,405	56,787
Total % change		−0.08	−0.23	−0.50	−0.37	−0.92	−1.99
% change in 5 high-emission industries	52,703	−0.08	−0.24	−0.51	−0.40	−0.99	−2.14
% change in the 25 other industries	5236	−0.06	−0.17	−0.38	−0.08	−0.21	−0.43

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
