# Peer review of "Beyond Strict Regulations to Achieve Environmental and Economic Health—An Optimal PM2.5 Mitigation Policy for Korea"

_ijerph, 2020, doi:10.3390/ijerph17165725_

Round 1
Reviewer 1 Report
Park et al. “Beyond Strict Regulations to Achieve Environmental and Economic Health - An Optimal PM2.5 Mitigation Policy for Korea”
Using regional static computable general equilibrium model, this study investigates the impacts on economy and environment by the air quality mitigation policies. The model selected two experimental regions and found a new optimal way by combining the higher tax rates and a tradable market for PM emission permits, which can reduce the PM2.5 emissions by 2.35% without sacrificing economic growth. While this is a model simulations study which makes the results highly dependent on the reliability/robustness of the model performance, this study provide a new policy paradigm for managing air pollutants. Shortly, I would recommend its acceptance for publication after necessary modifications.
Detail comments
Line 37-39, Not only in developed countries, but also in developing countries, the economic activities are heavily rely on the fossil fuel energy sources. Also, using “all” here is not suitable since other pollution sources exist. In addition, a few references is recommended here, such as Zheng et al. (2018, https://doi.org/10.1016/j.atmosenv.2018.06.029), Zhao et al. (2019, https://doi.org/10.1029/2018JD028888), and Fan et al. (2020, DOI: 10.1016/j.atmosenv.2019.117066).
Line 48-50, at the same time, the recent significant improve of air quality in China as indicated by Zhang et al. (2019, https://doi.org/10.1007/s13143-019-00125-w) and Fan et al. (2020, DOI: 10.1016/j.atmosenv.2019.117066) also highly benefits the atmospheric environment in Korea, which should be also mentioned.
Line 115, “emissions sources” -> “emission sources”
Figure 1. Latitude and longitude are necessary to readers for knowing the accurate locations. Explanation about the means of colored regions is helpful.
Figures 2 and 3. Information about the circle size and colors are necessary in the caption.
Figure 4. which year are the results for?
Line 184-187, more details about the nesting structure should be described.
Line 192-194, please rephrase this sentence: grammar error exists in “Equation (1) and (2) show balanced equations in the production sector are demand functions ...”
Figure 6. I would recommend a brief description about this figure in the main text.
Line 304-305. Sorry that I do not understand this sentence. What do you mean “while shutting down aging coal-fired power plants”?
Author Response
We appreciate your fast reply and valuable comments on the manuscript.
Attached please find our response to your comments.

Reviewer 2 Report
I suggest to reject the paper. In my opinion, the authors confuse the concept of emissions and concentrations of PM2.5. If one (as stated in various parts of the paper) wants to reduce PM2.5 concentrations, it is not sufficient to act on PM2.5 emissions, as in this way one is completely neglecting the secondary PM2.5 component of PM2.5 concentrations.
These issues of the difference between emissions and concentrations (and of the secondary PM2.5 component) cannot be neglected. It is not sufficient to build a system to reduce PM2.5 emissions, as PM2.5 concentrations depend also on NOX, NH3, SO2 emissions.
The general concept of the paper is interesting, but missing the link between emissions and concentrations (and missing secondary PM2.5) is a too important limitation, in my opinion.
Author Response
We appreciate your comments on the manuscript.
Attached please find our response to your comments.

Reviewer 3 Report
The authors presented PM2.5 reduction scenarios driven by different policy options using a computable general equilibrium model applied to Korea
divided in two regions considerening the actual different impact of pollutants.
The introduction is adequate to present the statement of the problem and the objective of the study is clearly defined.
However I think that some points have to be cleared:
- the authors cited in different parts of the text NOx and SOx emissions by fossil fuel power plants (i.e. lines 35, 132) and added discussion
about PM2.5. I think that the authors have to clear the relationship: e.g. did they mean that PM2.5 is mainly due to the formation of secondary
aerosol starting from SOx and NOx (which are gases)?
- A half of par. 2.2 is focused on fossil fuel power plants description and distribution in Korea, but in fig. 3 the color for energy production
(light blue) shows that only a little part of PM is produced by the above mentioned plants, moreover the main difference in percentage between
CN and the other part of the country seems to be the emisson from non-metallic mineral production and basic metal production. So, why focus the
chapter on power plants if there are other preminent sources of PM that were only listed and not described in detail? Or the authors refer to
the use of fossil fuels for powering operations in those different industries sectors? This part of the paragraph has to be cleared and
expanded.
- In fig. 3: Is the difference of magnitude of the pie charts proportional to the difference in total PM emission in the different regions?
- In fig. 7 (caption): what the authors mean with "domestic production"?
- Abstract: as the study has already been completed I suggest to use past tense
Author Response

(The authors gave the same response as above.)

Reviewer 4 Report
This manuscript uses a computable general equilibrium (CGE) model to investigate approaches other than strict regulations to mitigate the air quality issue in Korea. The authors suggest alternative strategies using tax rate and emission permit, which may reduce the PM2.5 emission without negative impact on the economic growth. The topic is interesting but this work seems to focus more on the policy side rather than the environmental pollution and public health. Thus I feel that this study does not fit the scope of this journal well. I think a major revision is required and more comments related to the contents could be found below.
Major comments:
- The overall structure of CGE model is unclear to me. I do not know what is the input and output of this model and how we could use it to link the emission control to economic growth.
- The results and discussion sections need to be improved to directly reflect the impact of a scenario on the economic growth.
- The authors propose some less aggressively and more economically efficient strategies. However, it is unclear to me that what is the impact of using the strict regulations proposed by Korean government on economy. The authors may use the CGE model or other way to evaluate the strict regulations and compare with the results in this work.
- The authors mention some future work related to health risk and pollutant emission in the conclusion section. These are important and valuable, and I think they should be explored at least to some extent so that this work is not just a hypothetical policy-based study.
Minor comments:
- Line 52: should 25.1 mg/m3 be ug/m3?
- Line 108: is the national PM2.5 goal calculated as annual mean of daily average?
- Line 140: the definition of short name “CN” has been made in the previous section.
- Line 147: why the GRDP of CN is only 6% though it plays an important role in the power generation?
- For Figure 2, it is unclear to me what the different sizes of pi chart mean. It is also unclear to me what the shaded colors mean as background
- Line 193: it may be helpful to explain the meaning of term “C” and “XDi”.
- Some terms in Equation (3) are not explained in the text.
- Linear 208: it is unclear to me the usage of permits. Can a polluter keep the current emission as long as it buys enough permits? What is free allocation of permits?
- For Figure 6, it will be better to highlight the input and output of CGE model.
- For Figure 7, is the high value of Basic metal products in CN due to its high emission or low domestic production?
- Add the descriptions to the short names in Table 2.
- Line 334: why the reduction of emissions in CN is higher than that in RP?
- Line 337: please provide more discussions about why a tradable market leads to a decrease of emission from 0.44% to 2.35%.
- Line 342: why the reduction is higher in RP than CN?
- Line 344: I did not find any result related to the cost of reducing PM air pollution.
Author Response
We appreciate your comments on the manuscript.
Attached please find our responses to your comments.

Round 2
Reviewer 2 Report
ok, I appreciate the changes done to clarify the emissions vs concentrations difference. I can now accept the paper for publication.
Author Response
Thanks for your valuable comments and suggestions.
Reviewer 3 Report
The authors have responded to my comments, I think that the manuscript has been improved, I appreciate that the limits of this study have been included in the discussion.
I still prefer past tense for the study description in the abstract
Please check English in capitions of figures 2 and 3.
Author Response
Thanks for your valuable comments and suggestion to improve this manuscript. We checked the English in Figure 2 and 3 as you suggested.
Reviewer 4 Report
It is better to merge the replies to the manuscript.
Author Response
Thanks for your valuable comments and suggestions to improve this manuscript.